# Experiences and perceptions of respectful maternity care among mothers during childbirth in health facilities of Eastern province of Rwanda: An appreciative inquiry

Alice Muhayimana[1,2], Irene Josephine Kearns[1]*, Darius Gishoma[2], Olive Tengera[2], Thierry Claudien Uhawenimana[2]

**1** Faculty of Health Sciences, Department of Nursing Education, School of Therapeutic Sciences, University of Witwatersrand, Johannesburg, South Africa, **2** School of Nursing and Midwifery, University of Rwanda, Kigali, Rwanda

* irene.kearns@wits.ac.za

**Data Availability Statement:** All data generated or analyzed during this study are included in this published article [and its supplementary

## Abstract

### Introduction

The World Health Organization (WHO) has emphasized the importance of ensuring respectful and dignified childbirth experiences. However, many countries, including Rwanda, have documented negative experiences during childbirth. Identifying best practices can help uncover sustainable solutions for resource-limited settings rather than focusing solely on the challenges and negative aspects. This study aimed to explore and describe how mothers in Rwanda's Eastern Province perceived and valued their childbirth experiences during their most recent labour and delivery.

### Methods

We conducted a qualitative, exploratory, descriptive study. Purposive sampling targeted mothers who reported receiving respectful care during labour and childbirth. We selected 30 mothers from five hospitals in the Eastern Province of Rwanda. Data collection involved in-depth interviews (IDIs) following the first four stages of the Appreciative Inquiry (AI) 5D cycle. We employed the thematic analysis and used NVivo 12 to organize codes and develop a codebook.

### Results

Three main themes, each with subthemes, emerged from the analysis. The first theme, *appreciated care*, included compassionate care and emotional support, autonomy and self-determination, timely care, privacy and confidentiality, and a supportive environment. The second theme *perceived respectful care*, addressed the meaning of respectful care and the sources of the participants' satisfaction. The third theme, *strategies for improvement*, focused on increasing women's self-control, sustaining providers' positive behaviours, and fostering caring leadership. Participants described receiving compassionate, empathetic,

information files]. The datasets generated and analyzed during the current study are available in the Excel output from Invivo 12 analysis of mothers' interviews repository.

**Funding:** The data collection of this research was funded by CARTA and University of Rwanda (UR)/ Swedish International Development Cooperation Agency (SIDA) program under Open grant/RMC/ 2022. The funders had no role in study design, data collection and analysis, decision to publish, or preparation of the manuscript.

**Competing interests:** No authors have any competing interests

and dignified care, with timely attention to their needs. They expressed satisfaction with the services provided, noting that healthcare providers were kind, polite, and dedicated, often going beyond their duties. Mothers recommended to be more involved in decision-making, maintaining an optimal environment for childbirth, and enhancing community trust and professionalism in maternity care. They also acknowledged their responsibility to ensure proper birth spacing.

## Conclusion

The aspects of care the participants appreciated could be consistently maintained and promoted. There is a need to build on the progress made in delivering RMC to strengthen community trust and elevate expectations. Given the sensitive nature of RMC, we recommend conducting further studies using the Appreciative Inquiry approach.

## Introduction

Respectful Maternity Care (RMC) is the optimal approach for providing care during the maternity period. RMC prioritizes preserving women's dignity, privacy, and confidentiality, preventing ill-treatment and abuse. RMC ensures autonomy, granting women the right to information and informed choice, which is recognized as a fundamental human right [1].

In 2018, the World Health Organization (WHO) emphasized that RMC extends beyond preventing morbidity and death, emphasizing the importance of experiencing a positive childbirth through the utmost respect and dignity [2]. Providing RMC contributes to positive childbirth experiences and reduces maternal and neonatal morbidity and mortality [2–4]. Respect is fundamental to Rwandan culture [5], influencing various traditional social interactions. Integrating respect into childbirth practices is crucial for upholding maternal and neonatal rights and overall well-being.

Mistreatment towards mothers during childbirth contributes to the poor quality of maternity care received by the mothers [6–8]. Disrespect and abuse during childbirth can cause long-term psychological harm and diminish women's confidence and self-esteem [7,9].Being disrespected may cause shame, sorrow, and insecurity; distrust and loss of confidence in the health care staff; a sense of powerlessness; and reluctance to call for help [10]. Studies reported that memories of the labour and childbirth experiences remain in women's minds for a lifetime and are repeatedly shared with other women [7]. Disrespectful and abusive care contributes to the fear of future utilization of the same maternity services, and reduced utilization of maternity services interferes with safe motherhood, contributing to an increase in adverse maternal and neonatal outcomes [7]. D&A is a violation of fundamental human rights.

In efforts to reduce maternal and neonatal deaths, significant progress has been made in Rwanda through various initiatives. Introducing trained Community Health Workers (CHWs) in 2009 was crucial in promoting maternal and newborn care at the household level [11,12]. In 2008, the country implemented a national health insurance policy, covering Rwandans with fees based on family income [13]. Presently, 91% of Rwandans have medical health insurance [14], resulting in a significant decrease in post-childbirth hospital detentions due to unpaid bills. The Ministry of Health prioritizes evidence-based practices, including Basic Emergency Obstetric and Newborn Care (BEmONC), skilled birth attendance, and family planning[15]. Critical programs involve CHWs identifying pregnant women [11], the

RapidSMS mHealth program facilitating timely transfers [12], and widespread ambulance availability [12]. The goal is to achieve 100% health insurance coverage shortly [13].

RMC has been integrated into BEmONC, with healthcare providers receiving RMC training and mentorship through programs such as the Maternal and Child Survival Program [16] and the More Happy Birthdays project [17]. National consultation meetings with RMC stakeholders were conducted in Rwanda to address the mitigation of Disrespect and Abuse [18]. These initiatives have been implemented in recent years in collaboration with the Rwanda Association of Midwives and the Ministry of Health, including the Maternal and Child Survival Program (MCSP) and the More Happy Birthdays project. Regardless of the progress made in Rwanda concerning RMC, the accomplishments are still less documented.

Despite the importance of RMC, previous studies have predominantly focused on negative experiences reported by mothers [6,8,19,20]. Little attention has been given to exploring the power of positive RMC practices. Instead, efforts to address maternity mistreatment and abuse have primarily been problem-solving oriented. However, there is a dearth of studies that examine RMC from a positive perspective [21]. In 2018, there was a shift towards promoting RMC instead of focusing on disrespectful and abusive practices during childbirth. To support this shift, WHO published a document on positive childbirth experiences [2]. Studies have emphasized the significance of compassionate interactions between healthcare providers and women. This includes providing timely and relevant care, using courteous language, and effectively exchanging essential information [21–24]. Teamwork among HCPs and their positive attitude towards mothers, good leadership, and a conducive environment with the necessary equipment and adequate physical setting lead to positive childbirth experiences [9,25–27].

AI in RMC is suitable as it focuses on improvement rather than blaming or taking punitive measures, making it less threatening for healthcare providers [21]. This approach appreciates achievements and fosters positive change [28]. AI approach is appropriate to our topic of focus (RMC) since RMC has a positive core. This approach encourages participants to reflect on their best experiences, fostering a more constructive and empowering perspective [5,28]. AI accentuates existing strengths, fostering a positive outlook that enriches relationships, culture, vision, learning, innovation, and collective endeavours [5]. This study systematically employed AI to address the gap in existing literature, encouraging mothers to share their positive experiences, and their insights serve as a foundation for advancing and maintaining RMC. Investigating the positive exemplary experiences and practices can identify feasible and culturally appropriate sustainable, valuable solutions within resource-constrained environments [21,29].

The absence of overt disrespect or abuse during childbirth doesn't necessarily imply that RMC practices are in place [30]. For example, just because there isn't any physical abuse doesn't guarantee the presence of compassionate and positive care behaviours [30]. RMC and mistreatment during childbirth represent opposite ends of the spectrum. Research shows that women and newborns can experience a blend of both positive RMC practices and negative mistreatment behaviours within this range [30]. Advocating for RMC and addressing mistreatment are simultaneous required efforts. Socio-economic and cultural contexts influence women's perceptions of RMC in each country. In research conducted on mothers, HCPs, and family members in Tanzania and Malawi using AI techniques, participants valued compassionate interactions and a conducive environment, including cleanliness, sufficient infrastructure and equipment, supportive leadership, and positive staff attitudes [21].

In Rwanda, no formal research has been conducted from both a women's rights perspective and a focus on positive birth experiences. Rosen and colleagues examined RMC based on the seven rights of women of childbearing age across five African countries, including Rwanda [31]. The findings highlighted systematic violations of women's rights in childbirth. Another study conducted in Rwanda by Mukamurigo and colleagues explored childbirth experiences

and reported on poor childbirth experiences [10]. Mothers were slapped, verbally berated, humiliated, reprimanded, physically assaulted, insulted, abandoned, or subjected to inappropriate and rough handling by healthcare providers [10]. A significant number reported being shouted at for failing to follow instructions, and some were retained within hospitals due to unsettled bills [10,31]. Rather than solely focusing on repairing damages or focusing only on challenges, there is a need to build upon positive experiences. Exploring and describing service users' experiences can maintain the existing appreciated RMC. Exploring and describing the existing appreciated RMC through the eyes and experiences of service users will embrace, establish, and maintain a culture of respect and dignity among healthcare providers towards mothers.

This research primarily explored how mothers in Rwanda's Eastern Province perceived and appreciated their childbirth experiences, specifically focusing on Respectful Maternity Care. This question is highly relevant, as understanding positive RMC experiences can offer valuable insights into effective practices and culturally appropriate solutions, particularly in resource-limited settings. The focus on positive experiences through Appreciative Inquiry (AI) added a unique dimension to the existing literature, which often emphasized negative aspects like disrespect and abuse.

## Methods

### Study design

This study is qualitative, descriptive, exploratory, and descriptive in nature. Its purpose is to describe and provide meaning to the world of appreciated RMC experienced by mothers during their recent childbirth within the respective health facilities. The researcher's focus was on identifying, portraying, exploring, and understanding the positive RMC experiences encountered by mothers during labour, childbirth, and the immediate postpartum period. The aim is to form a holistic picture of this phenomenon and gain comprehensive insights into the topic.

This study is part of an ongoing doctoral study that aimed to design strategies to sustain RMC in Rwanda. The entire doctoral research began with a quantitative survey on mothers [32], conducted simultaneously with interviews. Alongside data collection from mothers, we also gathered insights from HCPs at the same study sites to understand their perspectives on RMC as reported by the women [3]. Utilizing AI methods, we examined and described HCPs' views on RMC. Ultimately, strategies to sustain and promote RMC in Rwanda were developed and validated [33]. This current manuscript is only about the exploratory phase of the qualitative phase that aims to explore the RMC experiences received by mothers. This qualitative part utilized the Appreciative Inquiry (AI) method that guided the in-depth interviews (IDIs) conducted with mothers, aiming to gain a comprehensive understanding and knowledge of the broader perspective of RMC as experienced by mothers.

Given that RMC is a sensitive topic, using AI to explore this sensitive subject allows for focusing on the best experiences, discovering realistic and culturally accepted sustainable solutions, and generating constructive and innovative results within the constraints of limited resources [29]. AI emphasizes the power of positive traits and encourages individuals to consider strengths rather than blame [21,28,29]. The AI approach promotes positive thinking and enhances and sustains existing best practices rather than solely focusing on repairing harm. Exploring people's positive aspects enhances relationships and boosts positive emotions [29].

### Study setting and population

The study was conducted in hospitals located in the Eastern province of Rwanda. There are a total of ten hospitals in this province, and for the purpose of this study, five hospitals were

randomly selected. These included three district hospitals, one provincial hospital, and one referral hospital. The participants in this study were mothers who underwent labour and gave birth either through vaginal childbirth or caesarean section at the selected study sites and were being discharged.

## Ethics approval and consent to participate

All methods were carried out in accordance with relevant guidelines and regulations. The study protocols were approved by the University of Rwanda Institutional Review Board (Approval Notice: No 385/CMHS-IRB/2021), the National Health Research Committee (NHRC) of the Ministry of Health (MoH) of Rwanda (Reference: NHRC/2022/PROT/003), and the Human Ethical Research Committee (HREC) from Wits University (Approval Notice: No M220265). Permission was also obtained from the study sites. Informed consent was obtained from all participants involved in the study. None of the participants were minors.

## Sampling and data collection

We used purposive sampling to select 30 mothers recently delivered at the study sites. One of the inclusion criteria in this study was the mother who self-reported feeling respected during labour and childbirth. So, by recruiting them, the researcher wanted to know how they were respected. The Eastern province was selected because it is the second province with the highest birth rate based on the latest report from Rwanda 2019–2020 Demographic and Health Survey [34]. This study was concurrent with the other quantitative study about reported RMC [32], whereby mothers who reported feeling respected in general were invited to participate in this qualitative study. Before the interviews, the researcher purposively recruited mothers who had reported feeling generally respected. We recruited them after their discharge while their birth companions were dealing with the payment process. These mothers volunteered to participate in the study. The researcher provided them with detailed information about the study and the interview process. Participants who agreed to take part signed consent forms, agreeing to participate and to have their interviews recorded.

In-depth individual interviews were conducted with the mothers, taking place 12 hours after childbirth for those who had a normal childbirth and 36 hours for those who underwent a caesarean section because they were discharged from the hospital after that period according to the current treatment protocol. The Principal Investigator conducted the interviews in the local language (Kinyarwanda) and audio-recorded. The interviews were conducted in quiet and private areas within the hospital. Following the AI approach, the mothers were asked about their experiences and perceptions regarding RMC during labour and childbirth. The field notes were taken and were kept. Data collection took place from 1st to 30th July 2022. This study utilized semi-structured open-ended questions designed based on the first four stages of the 5D cycle of Appreciative Inquiry (AI), as Cooperrider, Whitney, and Stavros proposed [28]. These questions aimed to elicit responses from the participants (mothers) regarding their most positive experiences of RMC during labour and childbirth, as well as their future expectations for RMC.

**4D stages of AI and interview guide.** The **defining** stage of Appreciative Inquiry (AI) involves addressing and clarifying the topic of inquiry and ensuring it has a positive core[28]. Cooperrider and colleagues argue that understanding the positive core is vital when using the AI approach, emphasizing choosing affirmative topics that genuinely reflect what people want to explore or learn more about [28]. These topics focus on the positive moments and envision possibilities for a better future. The positive core is integral in all AI phases, highlighting and magnifying what gives the organization life [18,28,35]. In this phase of this study, the author

defined the topic of inquiry focused on exploring the best moments of RMC and the perceptions of mothers recently delivered in Eastern Rwanda's maternity facilities.

**The discovery** phase consists of identifying and appreciating existing positive aspects by exploring what works well. Questions that we posed centred around "What gives life?" or "The best of what is" to gather positive stories from participants. This phase aims to unveil the strengths and positive elements [18,28,36]. The discovery phase in AI involves identifying and appreciating the best aspects of experiences[18,28,36]. AI's unique feature is its positive framing of questions, focusing on peak moments of excellence[28,36]. The participants share stories of exceptional valued accomplishments, emphasizing core life-giving factors. Throughout the discovery phase, AI creates a supportive context for dialogue by maintaining a consistently positive orientation [28,36]. In the discovery stage, participants share their experiences, particularly their best moments [28,36].

In this study, we asked the mothers to reflect on their experiences and share their best moments of RMC/ their greatest experiences of recent labour and childbirth in the hospital. The specific question posed to the mothers was: *"What is the best RMC experience that you have encountered during your labour and childbirth in the hospital?" "At any point during labour and childbirth for this childbirth, tell me any other kind of treatment that made you feel good/dignified or respected? "What is the care were you happy with or loved the most during labour and childbirth?" "Why did you like that?"* The research team ended up identifying the vital themes from the stories together.

**The dream** phase consists of envisioning the overall vision, results, and impacts, as well as fostering innovation by asking questions like "What might be?" [28,29] This phase involves using the insights from the discovery phase to create a set of aspiration statements, guiding the design of future actions. Participants identify new possibilities for the future[5,35]. The dream phase envisions a positive future, building on qualities discovered in the Discovery phase[35]. AI grounds future images in the positive past, creating compelling possibilities from extraordinary moments [28]. The primary goals are fostering participants' dialogue by sharing positive stories and identifying common themes without critique [29,36]. AI does not focus on solving problems but emphasizes mutual discovery, guiding participants to envision an organization that embodies their hopes and dreams[5,35].In the dream stage, participants describe their dreams and desires and envision the possibilities of what could be considering their own needs and preferences [28,29]. The mothers and HCPs were asked to foresee and picture RMC for the future. In this phase, the study participants (mothers and HCPs) recommended sustaining and promoting the best-reported RMC practices.

In this study, the participants were encouraged to share their aspirations for RMC during their labour and birth process in the hospital. They were asked to think broadly and holistically about the future they desired. The specific question posed to the participants was: "*What would be the best RMC for you during your labour and birth process in the hospital? How do you wish it was?" "What do you think has made possible for your care to be good?"*

The **design** stage generates implementation options and co-constructs plans by focusing on the following: "What should be the ideal?"[5,29]. This phase aims to identify concrete actions to support the new possibilities envisioned in the dream phase[18,28]. Participants collaboratively create and commit to actions that will turn aspiration statements into reality [28,29]. The design phase articulates the strategic focus, including a vision of sustainability and a compelling statement of strategic intent[18]. The design stage gathers valuable insights from the participants to inform the development of practical steps and interventions that can enhance RMC within the hospital setting [29], focusing on recommending and developing strategies to sustain RMC within the hospital setting.

In this study, the mothers were asked to provide recommendations and suggestions to envision and portray an improved RMC for the future. The specific question posed to the mothers was: "*How should RMC be improved to contribute towards creating an RMC hospital environment?" "What can be recommended and developed to ensure the sustainability of RMC in the hospital?"*

## Data analysis

The research team transcribed the interviews verbatim in Kinyarwanda, and then the research assistant translated them into English. This approach helped to expedite the process while ensuring data integrity. Back translation was also performed by another research assistant and the research team as a quality check. The researcher read and reread the transcripts to gain immersion in the data. NVivo 12 was used for open coding and creating a codebook.

We employed thematic analysis. We followed a six-step process: familiarization, coding, generating themes, reviewing themes, defining and naming themes, and writing up [37,38]. We selected the supporting quotes to present the findings. The research team then engaged in multiple readings of the transcripts to become familiarised and immersed in the data. Line-by-line coding was employed to identify initial codes. The research team thoroughly reviewed the initial codes, seeking similarities and differences in participants' viewpoints. Through this process, categories were formed to group related codes. The research team carefully reviewed the identified categories, engaging in an iterative process that involved going back and forth between categories and the transcripts to ensure that participants' perceptions and experiences were accurately captured. The in-depth analysis of the identified categories by the research team allowed for identifying overarching themes, representing broader patterns and concepts within the data.

We applied evaluative criteria for rigour in qualitative research to uphold the quality and trustworthiness of the research findings, enhancing the overall validity and reliability of the study [37,39]. This rigorous analysis enabled the research team to derive meaningful insights and present the findings comprehensively and comprehensively[38]. The study ensured credibility by selecting women who self-reported being respected and employing open-ended interview questions. Sharing transcripts with mothers during data collection enhanced credibility. Dependability was ensured through meticulous decision documentation, while data stability was maintained with consistent and timely collection. Confirmability relied solely on mothers' narratives, avoiding researcher bias. The authors reflected on their personal biases and assumptions that could influence their research, starting from the formulation of the research question and continuing through to the conclusions drawn. The study's transferability was improved through a comprehensive study description using a well-structured interview guide [38].

## Results

### Participants' sociodemographic characteristics

The participants' sociodemographic characteristics included age, gravidity, number of living children, marital status, mode of childbirth, and status of the baby. The participants' ages ranged from 18 to 44 years, with a mean age of 26. The number of living children varied, ranging from zero to seven children. (**Table 1**).

The experiences and perceptions of mothers with regard to RMC related to three main themes and subthemes, namely, 1) Appreciated care with subthemes: compassionate care and emotional support, autonomy and self-determination, timely care, privacy and confidentiality preservation, and enabling environment; 2) Perceived respectful care with subthemes: the

**Table 1. Participant sociodemographic data.**

| Variables | Categories | Frequency *n = 30* | Percentage % |
|---|---|---|---|
| **Age** | 18–20 | 7 | 23.3 |
| | 21–35 | 20 | 66.7 |
| | 36 and above | 3 | 10 |
| **Gravida** | Primigravida | 12 | 40 |
| | Multigravida | 18 | 60 |
| **Level of the hospital** | District | 9 | 30 |
| | Provincial | 12 | 40 |
| | Referral | 9 | 30 |
| **Living children** | No child | 1 | 3.4 |
| | 1 child | 10 | 33.3 |
| | 2 children | 10 | 33.3 |
| | 3 children and more | 9 | 30 |
| **Marital status** | Cohabitated | 17 | 56.7 |
| | Married | 5 | 16.7 |
| | Single | 8 | 26.6 |
| **Mode of childbirth** | Vaginal birth | 20 | 66.7 |
| | Caesarian section | 10 | 33.3 |
| **Status of the baby** | Stays in neonatology | 4 | 13.3 |
| | Stays with mother | 25 | 83.3 |
| | Died | 1 | 3.4 |

meaning of respectful care and happiness motives 3) Suggested strategies with subthemes: women's self-control, providers' behaviours and caring leadership (Fig 1).

**1. Appreciated care.** Participants reported receiving compassionate care, free from mistreatment and violence. They expressed that their choices and preferences were respected throughout their recent childbirth experiences. They also reported receiving dignified and equal care, being attended to in a timely manner, and receiving the necessary services; their privacy and confidentiality were respected, and they received adequate information regarding their health status and care.

*Compassionate care and emotional support.* The participants expressed examples of feeling treated with empathy and compassion. These included: receiving anaesthesia before suturing vaginal tears and episiotomy, being provided with painkillers after caesarean birth, and being offered support to manage labour pain. Participants who gave birth by caesarean section were systematically provided with painkillers after childbirth. Those who experienced pain during normal childbirth were comforted; however, they were also aware that pain is a natural part of the childbirth process and that there were no specific interventions to reduce pain. Mothers expressed that although they had to endure the pain, HCPs should address all of a mother's concerns during childbirth.

*"I don't know how I can describe that care I received here; it's too much. After the operation, they start giving me perfusion and giving me pills to reduce pain. . ."It is the first time I gave birth; the first thing that ever causes pain is labour. However, when someone cares about you and ensures you that labour will progress like that or that, you feel energized."* **Participant 12, aged 28 years, gravida 1, C/s** childbirth

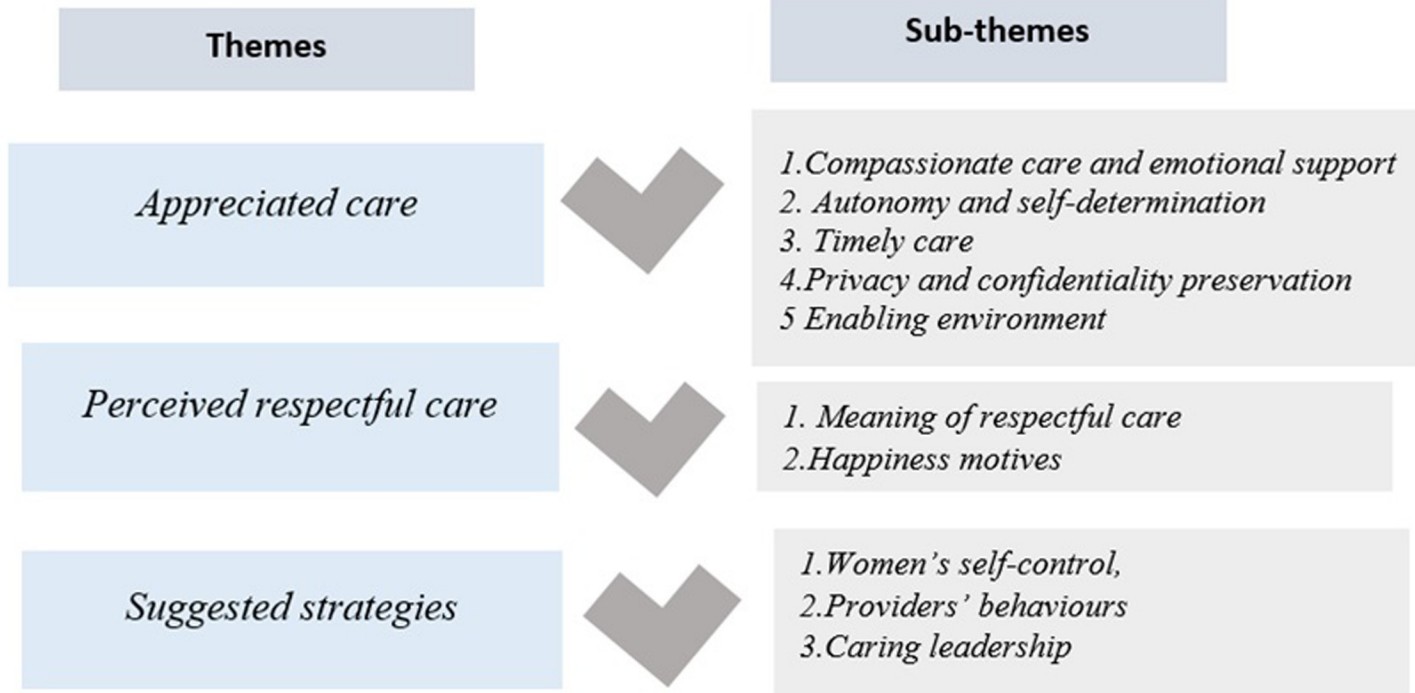

**Fig 1. Themes and sub-themes.**

*"I heard that pain is necessary; you have to suffer and get well. . . our parents tell us that you have to suffer, but you will be fine. . .Because there is nothing to reduce the pain, they (HCPs) are saying be patient; you will get better."* **Participant 26, aged 22, gravida 1, normal childbirth**

Mothers appreciated the dignified care they received and consistently said that they felt happy and satisfied with the services they received. The HCPs were patient with the mothers and spoke to them with comforting words, kindness, and politeness. The mothers explained that they had not been harassed in any way, that the HCPs answered their questions without rudeness, and used gentle and soft expressions when interacting with them. Mothers were grateful because they felt valued and that the birth attendant had stayed close to them throughout the process. It meant a lot when the birth attendant praised them for pushing the baby out because it gave the courage. The mothers described the staff as lovable due to their kindness and caring attitude.

*"In healthcare providers, there was a lady who told me ", mama, push, be patient, be patient darling, push; the contraction is in. I felt happy". . . the midwife comforted me and told me: 'all the tears you cried, I want to wipe them away'. . .. Now I am satisfied, I don't know how I can say it, but it made me very happy"* **Participant 17, aged 26 years, gravida 2, normal childbirth.**

*Autonomy and Self-determination.* Mothers reported feeling freedom during their childbirth experiences. They were not restricted from praying and were allowed to eat and drink; they had the freedom to walk, although within a limited space. The participants received equal treatment, there was no favouritism, and the hospital assisted those in need, such as providing

diapers for the baby. Mothers further stated that their choices and preferences were respected and felt supported by the HCPs. Mothers who came without a transfer letter were also received with kindness, and the participants were not denied any service and received assistance every time.

*"I told them my choice for family planning, and they immediately did it to me. I told them that I had decided to use the sterilization, and I had to talk to the doctor in charge, and he did it for me."* **Participant 28, aged 40, gravida 8, normal childbirth**

Mothers reported that they received guidance on how to behave well during their childbirth experiences, including being advised on precautions to take and practising self-control. They expressed happiness at being shown the placenta and receiving information about family planning and birth spacing. Some mothers were taught breastfeeding techniques and were encouraged to have regular meals to ensure an adequate milk supply. Additionally, some mothers were informed about the progress of their labour and encouraged to stay hydrated by taking fluids.

*"Every time I needed the doctor, he came and told me that you are on this point, you are going to give birth well, relax. When I needed him again, I called him until I gave birth, and he assisted me well, comforting me, telling me to push, and do it quickly without being rude to me but with kindness".* **Participant 21, aged 21 years, gravida 2, vaginal childbirth**

Mothers were happy simply because they were not being coerced, and they expected that consent would not happen. They were informed and requested to provide informed consent, especially those who experienced caesarean section.

*"Doctor always asked me " Would you come for a check-up to see your labour progress? Would you come for an ultrasound to check if the baby is still alive? and I said yes. They told me that I am going to have surgery, they told me that you have to sign and your companion needs to sign too; once you have signed, you have agreed to the surgery they are going to do for you".* **Participant 13, aged 28, gravida 1, C/s childbirth.**

However, most participants were not explicitly asked for verbal consent during routine examinations. Mothers reported that they did not feel the need to be asked for permission as they trusted the expertise and abilities of the HCP. Mothers believed that it was their responsibility to follow the instructions given by the HCP. Nonetheless, participants expressed satisfaction with the practice of negotiation instead of coercion.

*"When they take you to the examination bed, you are already aware that they are going to examine you…. well, when we are there, there is no need for permission because the doctors do what they were taught, or they were appointed for. Whatever they do, you feel it is right… Listen, because of pain, you don't have to argue; you even think the service provider is the one to recover you."* **Participant 18, aged 23 years, gravida 2, vaginal childbirth.**

*Timely care provision*. Most of the mothers expressed happiness and were impressed by the care received. They reported receiving the best services and all the necessary care. The HCP attended to them promptly and assisted them every time, even when they were working overtime. In some hospitals, they first use available medicines, and mothers reimburse them later. The entire process was well organized, and mothers who arrived in critical conditions were

given hope and excellent care. When the HCPs could not manage a case, they called experts for assistance. One notable example was when a very experienced midwife successfully helped a mother deliver vaginally despite indications for a caesarean section.

*"The midwife gave me the best possible care. He was a good-hearted person, and I feel very happy. I was so happy with what they did to me that I felt like giving her a car; poor me, I don't have it. . . . Whenever I wanted the doctor, he came to me every time I called him, and he did not complain no matter how much I disturbed him"* **Participant 21, aged 21 years, gravida 2, vaginal childbirth.**

*"The thing that made me happy was that I had weak contractions the whole day, and they gave me a pill to boost them, and I got strong contractions. They also did an ultrasound and told me that the baby was in a good position, and I would give birth without any problem.*" **Participant 10, aged 22, gravida 1, vaginal childbirth**

*Privacy/Confidentiality preservation.* The participants described privacy as an important part of feeling well cared for. Privacy was ensured as described by the mothers. They reported that they were not exposed to the public during the childbirth process, and curtains or doors were closed to maintain their privacy. Mothers were allowed to be alone with the HCP, and cleaners were not allowed to enter the room during their stay.

*"I am thankful that these cleaners cleaned after you gave birth; no one ever saw me. In the* childbirth *ward, there are large curtains. Even the cleaner who brought the materials, the doctor took them without letting her in. . .. When a person wanted something, the service provider went out and opened the curtains to listen to what he needed. "* **Participant 5, aged 27, gravida 2, vaginal childbirth.**

To ensure confidentiality, healthcare providers spoke quietly to the mothers. However, mothers reported that they could hear other mothers' health information. Nevertheless, mothers revealed that it did not matter to them, as they shared suffering from labour pain.

*"Even though sometimes, we are many here, I can hear the health information of my peers and vice versa, but since you are all suffering, it doesn't matter. . .it would not matter because a other women in the hospital can't share my information knowing how herself she was doing in labour and the way we shared sorrow of labour pain."* **Participant 18, aged 23 years, gravida 2, vaginal childbirth.**

*Enabling environment.* Some women were excited to give birth in a large building (on an upper floor). However, mothers reported that the birth companions were not allowed to stay with them in the waiting room, particularly in the childbirth rooms, due to the structure of maternity rooms. However, most participants believe that birth companions are not helpful during the second stage of labour and that only HCPs are of assistance. In the Rwandan context, the birth companions are actually family members such as the mother of the woman, her mother-in-law, sisters or friends. The participants narrated that birth companions are not health professionals and can become scared and discourage the mother. Participants also reported that birth companions are often busy buying medicines or doing other outside tasks. They suggested they should simply stay nearby and be available to bring the baby's clothes after birth.

*"The birth companion could not be of help; she wasn't a doctor to examine me; she was only responsible for standing there near the door and waiting for them to say that I gave birth and brought the baby's and mother's clothes. I would not want my birth companion to come in! because even now, birth companions can see how you are doing and get scared and say that you are dying, but a healthcare provider, even if you are going to die, keeps telling you to be patient."* **Participant 26, aged 22, gravida 1, normal childbirth.**

Some mothers expressed satisfaction with the overall hygiene and cleanliness of the hospital. Cleaned toilets and regularly changed bedsheets made them feel satisfied. HCP often requests the cleaners to clean rooms multiple times. However, while some mothers were pleased with hygiene, others complained that the cleanliness in the toilets was insufficient.

*"I am also happy that the hospital is clean, you sleep, and they take care of your sheets and change them for you, and the shower and toilet are all clean without any problems. There was no problem, and the cleanliness was satisfying."* **Participant 28, aged 40 years, gravida 8, vaginal childbirth.**

**2. Perceived respectful care.** Participants expressed their understanding of respect during labour and delineated the elements that contributed to their contentment with the services provided.

*Meaning of respectful care.* Most participants understand that RMC encompasses privacy, confidentiality, and freedom of movement during labour. They associate RMC with receiving timely and compassionate care, being spoken to in a soft and compassionate tone, being spoken kindly and politely, comforting her, actively listening, and staying close to her. RMC is seen as attending to the mother promptly, starting with those in critical condition. It involves receiving the mother well, showing patience, and inquiring about her well-being and that of her baby rather than abandoning, harassing, ignoring, or silencing her.

*"The respect is when health service does not expose you in public if you arrive at the hospital, you are cared for, when a service provider comforts you by saying, sorry mom, sorry darling, with a smiling face, that is good. . ... When you are in the room, you are in a private place; you are private if there is a door, they put on a curtain or close it, and you feel that you are alone with a healthcare provider, and you feel you are respected."* **Participant 17, aged 26 years, gravida 2, normal childbirth.**

*"I feel that respecting a mother means being there when she needs you. You should make her feel comfortable and free to express herself. A mother should be allowed to move around during labour because it helps her feel better."* **Participant 21, aged 26 years, gravida 2, normal childbirth.**

Mothers perceive professionalism as a crucial factor in providing RMC. They value love, compassion, and a sense of being valued by HCPs as important facilitators of RMC provision.

*"What makes service providers give me good care is the love and compassion healthcare providers have or that they value their duty. Maybe I had met doctors who knew what they wanted to do or who valued their work ".* **Participant 18, aged 23 years, gravida 2, vaginal delivery.**

*Happiness motives.* Mothers expressed that fulfilling their desires was the most important reason for appreciating and feeling proud of the services they received. Their primary wish and expectation were to have a normal childbirth and a healthy baby while maintaining their own health. Additionally, mothers reported negative anticipatory feelings when seeking maternity care. However, HCPs were available round the clock, closely monitoring and encouraging them and instilling hope. Mothers appreciated that HCPs were not preoccupied with phones, treated them with respect, offered comfort and support, and showed genuine concern for their well-being. One mother even expressed happiness at reuniting with a midwife who had assisted her in a previous childbirth fifteen years ago. The perceived drivers of RMC, as reported by the mothers, included displaying love and professionalism, having a sense of responsibility, valuing both the work, receiving appropriate training, and the mother's critical state. The moment of getting their newborn baby was described as a transformative experience, overshadowing the pain of labour.

*"It was truly great. The doctor was telling me to relax and not worry; there was no other reason; service providers told me that I wanted to have a healthy baby because the doctor told me, "I will help you to give birth well. I did not have surgery. The thing that I am happiest about is that I saw a healthy child. For me, the first thing that makes me happy is to see a healthy child."* **Participant 11, aged 28 years, gravida 3, normal childbirth.**

Remarkably, one participant who had experienced the loss of their baby due to severe prematurity expressed appreciation for the services received.

*"From when I arrived until I left down there, up to now, healthcare providers took good care of me though my baby passed away. If I were able to reward them, I would do it; I would repay them if I had something. They took care of me".* **Participant 24, aged 20, gravida 1, C/s childbirth.**

Very few participants had positive expectations, such as hoping for a quick delivery or expecting to receive an ultrasound examination. Since most of the mothers were referred, some also anticipated a smooth and uncomplicated childbirth, and all of these expectations were fulfilled.

*"I saw that my baby was still alive in the womb. . .They just showed me her and told me she is a girl and ensured me that she is healthy; that makes me happy".* **Participant 13, aged 28 years, gravida 1, C/s childbirth.**

Participants generally reported low expectations when seeking maternity healthcare due to the information received from their communities. Mothers expressed concerns upon going to the hospital for childbirth, having heard negative stories from others and anticipating potential insults and physical mistreatment. Some were unaware of the current procedures and expected to receive the same treatment as in previous years. They anticipated being physically abused, insulted, scolded, subjected to fundal pressure, or unnecessarily undergoing a caesarean section. They also expected to be exposed naked in public and rudely spoken. They came with the fear of dying, being provoked, ignored, and being required to pay before receiving any service. However, participants experienced positive experiences.

*"I was thinking that I get here and give birth by insulting me; they don't hit me as I have been told. Before coming here, I have been told that some are beaten; I thank God they did not beat*

*me. I found it different; I've met a kind doctor. . . And the fact that I arrived here, they cared for me, they did not beat me, and then I gave birth immediately, that's made me happy".* **Participant 16, aged 19 years, gravida 1, vaginal childbirth.**

**3. Suggested strategies.** To sustain RMC, mothers provided recommendations to service users, caregivers, and hospital management. Each level has its own responsibilities to fulfil.

*Mother Self-control.* Mothers reported that they have a responsibility to respect HCPs by following the instructions given by HCPs, using family planning to space their births and regain confidence. To ensure the lifelong sustainability of RMC, they wish that their children would attend school and become future HCPs dedicated to providing quality services.

*"Mothers should respect health care providers too; we should have few children. You know, mothers, we always give birth, and you find that it is stressful, and we have to plan for the family".* **Participant 30, aged 29 years, gravida 3, vaginal childbirth.**

*Provider behaviours.* Respondents advised the service providers to sustain their courtesy, respect, and love towards the mothers. They should ensure the mothers' comfort and feel responsible in providing maternity care, providing guidance to the mothers and maintaining the existing culture of timely care, mutual understanding, and respect. The mothers advised the providers to re-evaluate themselves, work diligently, and care for the mother as a human being. The mothers recommended that doctors respect subordinates, such as cleaners, and cooperate with midwives, nurses, and mothers.

Mothers requested HCPs to prescribe affordable medicine and avoid unnecessary prescriptions of surplus medical supplies. Providers should explain the progress of labour and communicate the results of examinations to the mother. It was suggested that HCPs talk with the mother while providing care, explain the labour process and progress, and assist mothers in having a normal childbirth as much as possible. Additionally, it was advised that HCPs prioritize triage and handle emergency cases quickly. Mocking multigravida women and unnecessarily retaining mothers in the hospital were discouraged. Security personnel at the gate were also advised to be briefed on respecting and allowing mothers to enter the hospital.

*"Doctors should take care of patients as if it is their responsibility, not because it is their job, but because it is their responsibility".* **Participant 11, aged 28 years, gravida 3, normal childbirth.**

HCPs should strive to understand each other and avoid quarrels between them. One mother reported hearing HCPs insulting one another, while other HCPs complained about working overtime. Mothers advised that such behaviour should be stopped, as the consequences could ultimately affect the service users.

*"There is a time a healthcare provider come and found her fellow did not fill the form accurately then insulted her. . . I hate insulting . . .. some healthcare providers were complaining that there are some who work days and nights, saying a lot of things like, "Why didn't you give me my off time?" Healthcare providers came saying, "I am fed up with this. I may leave this job. Look, I did a day shift, and I did not take a break. They want me to fill out these while my working hours were finished.". . ., if a health care provider claimed to her colleague that she is tired, you could listen to her".* **Participant 24, aged 20 years, gravida 1, C/s childbirth.**

*Caring leadership*. Mothers recommended that the hospital leadership train, supervise, and innovate ways to motivate the HCPs. Participants emphasized the importance of publicly showing gratitude to the HCP and ensuring timely payment of salaries. Increasing the number of HCPs and providing sufficient labour monitoring equipment are suggested. Mothers also highlighted the need for continuous training for HCPs, including new graduates and students, and providing mentorship to ensure the childbirth of quality care. They recommended that future HCPs be taught to prioritize patient care and love for their patients. Conducting campaigns and workshops on RMC was seen as beneficial. Another important suggestion was to build or renovate hospitals with spacious rooms and an adequate number of beds to avoid mothers and newborns sharing beds. Last, they emphasized the importance of striving to have a good reputation in the community.

*"Doctors leaders would come to their colleagues and perhaps thank them and tell them that maybe all mothers who gave birth in this certain time are happy, and I think it will motivate them to continue doing well".* **Participant 15, aged 32 years, gravida 3, vaginal delivery.**

Some respondents recommended that the hospital provide food to mothers whenever possible, especially for those who come from a distance and may not have money to buy food from the canteen. Regarding hospital bill payments, it was suggested that priority should be given to providing care to the mother first, with payment arrangements made afterwards. They also recommended that the cost of services should be proportionate to the service received and that prescribed medications should be covered by health insurance. Respondents suggested that the hospital and the mother could enter into a contract allowing for gradual payment to address the issue of mothers being retained in the hospital due to an inability to pay. Participants emphasized the need to improve hygiene and cleanliness in the hospital environment. Regarding personal hygiene, some mothers expressed resistance to using diapers during labour, fearing harm to their babies. Mothers also raised concerns about hospital overcrowding, where two mothers may have to share one bed.

*"Another thing is that they should add hygiene and cleanliness, and they should clean properly so that mothers will not acquire illness.*" **Participant 16, aged 28 years, gravida 3, normal chilbrith.**

## Discussion

Our findings indicate that there has been significant progress in RMC in Rwanda over the last few years. The participants expressed excitement about the respectful care received. Mothers narrated to receive care with compassion, empathy, dignity, equality, and privacy. Mothers reported being attended to in a timely manner and receiving the necessary services. They mentioned signing consent forms and receiving guidance on how to behave during their stay. Mothers reported that HCPs used comforting words and displayed kindness, politeness, hope, love, and humanism. The professionalism of the HCPs was greatly appreciated, as was their sense of responsibility and the value they placed on their work, even going the extra mile by working overtime to save mothers and babies. Overall, the participants' desires were met, and they expressed satisfaction. These findings align with global evidence on RMC, highlighting the importance of emotional support, positive attitudes of healthcare providers, and effective communication during labour and childbirth [4,21,24,40,41].

Although our participants expressed appreciation for almost every aspect of the care they received, there are areas that we believe require improvement. Expressly, pain management

during normal childbirth was limited to emotional support, and the mothers were aware that pain is a necessary part of giving birth. The literature argues about other relaxation techniques and painkiller medications that could be utilized to alleviate labour pain and their effectiveness [42]. It was too emotive to discover that most of the mothers attended health facilities with low expectations, but fortunately, they had a positive experience and received more care than they anticipated. Participants suggested that the HCPs could explain to the community that the services provided in maternity have been greatly improved compared to the past. This finding is similar to the recommendation from the interventional study conducted in Kenya, which suggested using open maternity days to engage the community in maternity services [22]. Mainly, mothers emphasized that what is important to them is to have a healthy baby and to stay healthy. However, the WHO in 2018 emphasized that RMC goes beyond merely reducing maternal and neonatal mortality; it also aims to ensure a positive childbirth experience [2]. The RMC scoping review published by Jolivet in 2021 indicated that, globally, many women experience a mix of both positive and negative childbirth experiences. The review recommended promoting RMC while addressing and reducing disrespect and abuse [30].

Participants recognized the improvements made in maternity buildings, and mothers expressed excitement about giving birth on the upper floor of a new and beautiful building at most health facilities. However, there are still some inconvenient wards within the facilities. Mothers often labour together in open rooms or rooms separated by curtains, leading to the unintentional breach of confidentiality principles. Interestingly, mothers reported that they are not offended by hearing each other's health information since they are all suffering and going through similar experiences. Studies have revealed that inadequate labouring environments are a common issue in many low-income countries' healthcare settings, and labour wards are open rooms [43,44]. Consequently, the rooms cannot accommodate birth companions[44,45]. A facilitating environment with the necessary equipment and adequate physical setting leads to positive childbirth experiences[21,29]. A positive aspect of our study sites is that most of the maternity buildings have been recently renovated, and the remaining ones are scheduled for renovation in the coming days. Improving infrastructure may be perceived as challenging to implement RMC and could take time. However, HCPs can work with existing resources, such as using partitions and curtains, to maintain mothers' privacy [21]. Surprisingly, mothers mentioned that birth companions are not necessary while pushing the baby; they believe that birth companions are not of help at that stage, and the HCP are the only ones needed at that particular moment. However, evidence demonstrates that birth companionship improves RMC [9,24,46,47].

Mothers believed that there was no need for HCPs to request them to consent for examinations and routine care, and mothers were not disturbed by that because they fully trusted HCPs' knowledge and skills. However, consent for care, either written or oral, is always necessary [2,48]. Participants highlighted interventions such as HCP's training, supervision, and mentorship for new graduates as key factors in sustaining and improving RMC. These findings align with previous interventional studies conducted in the field [22,47,49,50] that have demonstrated the effectiveness of such interventions in RMC promotion.

In Rwandan culture, it is expected to show respect to pregnant women and help the woman in labour. The woman's and in-law's families normally accompany her to the hospital when she goes for delivery. This care is typically provided by her own mother, grandmothers, mothers-in-law, and the in-laws' family, as well as the broader community [51].

When a mother delivers a newborn baby that is alive and healthy, both her family and her husband's family pay a visit to express their congratulations and celebrate the occasion with various gifts, particularly food items, to the family. This is commonly referred to as' to reward the mother' or "guhemba umubyeyi" [51]. In the maternity services at the study sites, birth

companions typically stay outside the childbirth room or inside the labour room, not in the childbirth rooms, and their particular role is to make payments and bring medicines, food and clothes for the baby and the mother. It is also worth noting in this study that some women in the study prefer not to have their birth companions stay with them in the delivery room by saying that having birth companions in the childbirth room is unnecessary. Women may want to maintain their privacy during delivery and childbirth.

## Conclusion and recommendations

In this study, the mothers reported receiving compassionate, empathetic, dignified care and necessary services, and they were attended in a timely manner. Participants were satisfied, and their needs were met. Participants stated that healthcare providers were kind, polite, and dedicated, often working overtime. The participants appreciated compassionate care with emotional support, autonomy and self-determination, timely care, privacy and confidentiality, preservation, and an enabling environment. The participants explained how they perceived RMC. They suggested strategies to increase women's self-control, sustain positive providers' behaviours, and caring leadership. Mothers recommended including them in decision-making, maintaining an optimal childbirth environment, and increasing community trust and professionalism. Mothers also recognize their responsibility for birth spacing.

This study emphasizes the importance of maintaining and promoting RMC to provide women with care centred around their needs and a positive childbirth experience. The study suggests that healthcare providers should provide timely, compassionate, and emotionally supportive care to improve an environment that maintains privacy and confidentiality, fosters clear and effective communication, and empowers mothers' autonomy, satisfaction, and self-determination. Additionally, the study reveals a need to maintain and promote an adequate environment around facility childbirth to preserve and uphold community trust towards maternity services. Healthcare providers should pursue professionalism, ethics, and supportive leadership to sustain RMC. Mothers reported that they should contribute to being provided RMC by birth spacing.

Some steps have been taken to improve RMC in Rwanda, such as RMC training for healthcare providers in hospitals, but these still need to be documented. Mothers expressed their appreciation for the care received despite initially having low expectations when visiting the hospital. Raising public awareness about the universal rights of women and newborns will increase expectations and trust among mothers seeking maternity care in Rwanda. Mothers' primary concern was having a healthy baby and maintaining their health. While they acknowledged infrastructure improvements, they also emphasized the need for further enhancements. The participants suggested strengthening labour support by providing access to labour pain medications, enhancing confidentiality, and improving infrastructure and amenities. This includes constructing more spacious rooms separated by walls rather than curtains, where possible. Educating non-healthcare personnel, such as cleaners and security officers, is also essential for respecting and supporting mothers. It is recommended that communities be informed that healthcare providers' behaviours have improved and that more studies be conducted using AI approaches, as RMC is a sensitive topic [20,21] with a positive core.

## Strengths and limitations

### Strengths

AI is an innovative approach that proved effective in understanding the specifics of what respectful care looks like in Rwandan culture, enabling the gathering of enough information and encouraging mothers to share their appreciated RMC experiences. AI is less threatening

and, therefore, particularly suitable for RMC as we focused on the best aspects in this study. AI offered the participants genuine commitment, energy and motivation to discuss their experiences [28,29] openly. AI is engaging, powerful, and inspiring, focuses on the positive aspects of situations [29] and can potentially reveal creative solutions and ideas that may not emerge solely from addressing disrespect and abuse. The recruitment of participants who reported being respected in general allowed for collecting valuable data on their positive experiences. To the best of the author's knowledge, no previous studies on RMC conducted in Rwanda have utilized AI methods.

## Limitations

Although Appreciative Inquiry has its benefits, it also has its limitations. Prioritizing positive narratives and experiences during the discovery phase potentially dismisses participants' negative experiences. This may impede essential and meaningful conversations [29]. All parties involved must be genuinely committed and willing to concentrate on the positive aspects, even in challenging situations[29]. While innovative, the use of the AI approach might introduce some bias towards positive experiences. Including data on negative experiences or challenges faced, even within the AI framework, could enhance the credibility and balance of the findings. AI can be challenging to transition for people who are used to problem-solving approaches [29]. AI may not be suitable for some circumstances, especially when immediate action is needed, and a problem-solving approach may be more appropriate because AI necessitates persistence, practice, and patience [29].

The other limitation was that the study was conducted at the hospital exit time when the mothers were still on the hospital premises; this can lead to desirability bias. To minimize this concern, participants were assured of the complete confidentiality of their information, and interviews were conducted in a private room. It is important to note that mothers' experiences of RMC are subjective and can be influenced by socio-cultural factors. In addition, it is impossible to know how representative our participants are of Rwandan women's experiences.

## What is already known on this topic?

In the research conducted in Rwanda in the past, women emphasized the negative experiences of women during labour and childbirth. Women described being slapped by providers and spoken to rudely as concerning. Women reported experiences of humiliation, scolding, dehumanization, being tied to the childbirth bed, pinched, pushed, insulted, shamed, abandoned, and subjected to inappropriate and rough touching. Many women mentioned being yelled at for not complying with instructions, and some faced stigmatization, abandonment, and even gave birth on the floor. Additionally, those who failed to pay were reportedly detained inside hospitals [10,42]. However, what is not known is the perceptions and experiences of positive experiences of women during labour and childbirth.

## What this study adds

This study highlights the positive aspects of Respectful Maternity Care experiences among women in Rwanda, whereas previous RMC research in the country has primarily focused on the negative aspects [10,42]. Rwandan women in our study reported receiving care with compassion, empathy, dignity, equality, and privacy, and they were satisfied. Mothers noted timely attendance, providing necessary services and guidance on behaviour during their stay. They also reported that HCPs used comforting words and displayed kindness, politeness, hope, love, and humanism. HCPs were greatly appreciated for their sense of responsibility and dedication, even going the extra mile by working overtime to save mothers and babies. Participants

emphasized the need to be included in the decision-making process, maintaining an optimal environment for facility childbirth, upholding community trust, and supporting healthcare professionals in maintaining professionalism. Mothers acknowledged their responsibility, particularly in terms of birth spacing.

## Supporting information

**S1 File. Excel output from Invivo 12 analysis of mothers interviews.**
(XLSX)

**S2 File. Inclusivity-in-global-research-questionnaire.**
(DOCX)

**S3 File. PLOSOne human subjects research checklist (1).**
(DOCX)

## Acknowledgments

We would like to thank the women who participated in this study. A vote of thanks goes to the hospital managers of the study sites and to the data collectors.

## Author Contributions

**Conceptualization:** Alice Muhayimana, Irene Josephine Kearns.

**Data curation:** Alice Muhayimana, Irene Josephine Kearns.

**Formal analysis:** Alice Muhayimana, Irene Josephine Kearns.

**Funding acquisition:** Alice Muhayimana.

**Investigation:** Alice Muhayimana.

**Methodology:** Alice Muhayimana, Irene Josephine Kearns, Darius Gishoma, Thierry Claudien Uhawenimana.

**Project administration:** Alice Muhayimana, Irene Josephine Kearns.

**Resources:** Alice Muhayimana, Olive Tengera.

**Software:** Alice Muhayimana.

**Supervision:** Irene Josephine Kearns.

**Visualization:** Alice Muhayimana.

**Writing – original draft:** Alice Muhayimana, Irene Josephine Kearns.

**Writing – review & editing:** Alice Muhayimana, Irene Josephine Kearns, Darius Gishoma, Olive Tengera, Thierry Claudien Uhawenimana.

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
