## [Decision Letter · Decision Letter 0]

30 Sep 2024

PONE-D-24-26291Experiences and perceptions of respectful maternity care among mothers during childbirth in health facilities of Eastern province of Rwanda: An Appreciative InquiryPLOS ONE

Dear Dr. Muhayimana,

Thank you for submitting your manuscript to PLOS ONE. After careful consideration, we feel that it has merit but does not fully meet PLOS ONE’s publication criteria as it currently stands. Therefore, we invite you to submit a revised version of the manuscript that addresses the points raised during the review process.

We look forward to receiving your revised manuscript.

Kind regards,

Mukhtiar Baig, Ph.D.

Academic Editor

PLOS ONE

Journal Requirements: When submitting your revision, we need you to address these additional requirements. 1. Please ensure that your manuscript meets PLOS ONE's style requirements, including those for file naming. The PLOS ONE style templates can be found at https://journals.plos.org/plosone/s/file?id=wjVg/PLOSOne_formatting_sample_main_body.pdf and https://journals.plos.org/plosone/s/file?id=ba62/PLOSOne_formatting_sample_title_authors_affiliations.pdf 2. Please include a complete copy of PLOS’ questionnaire on inclusivity in global research in your revised manuscript. Our policy for research in this area aims to improve transparency in the reporting of research performed outside of researchers’ own country or community. The policy applies to researchers who have travelled to a different country to conduct research, research with Indigenous populations or their lands, and research on cultural artefacts. The questionnaire can also be requested at the journal’s discretion for any other submissions, even if these conditions are not met. Please find more information on the policy and a link to download a blank copy of the questionnaire here: https://journals.plos.org/plosone/s/best-practices-in-research-reporting. Please upload a completed version of your questionnaire as Supporting Information when you resubmit your manuscript. 3. We note that the grant information you provided in the ‘Funding Information’ and ‘Financial Disclosure’ sections do not match.  When you resubmit, please ensure that you provide the correct grant numbers for the awards you received for your study in the ‘Funding Information’ section. 4. Thank you for stating the following in the Acknowledgments Section of your manuscript: "We would like to thank the women who participated in this study. A vote of thanks goes to the hospital managers of the study sites and to the data collectors. We would also like to thank our study partners, CARTA and facilitators and the UR/SIDA program for providing funds for this work and Rwanda MoH." We note that you have provided funding information that is not currently declared in your Funding Statement. However, funding information should not appear in the Acknowledgments section or other areas of your manuscript. We will only publish funding information present in the Funding Statement section of the online submission form. Please remove any funding-related text from the manuscript and let us know how you would like to update your Funding Statement. Currently, your Funding Statement reads as follows: "The author(s) received no specific funding for this work" Please include your amended statements within your cover letter; we will change the online submission form on your behalf. 5. Thank you for stating the following financial disclosure: "This research was funded by CARTA and University of Rwanda (UR)/Swedish International Development Cooperation Agency (SIDA) program." Please state what role the funders took in the study.  If the funders had no role, please state: ""The funders had no role in study design, data collection and analysis, decision to publish, or preparation of the manuscript."" If this statement is not correct you must amend it as needed. Please include this amended Role of Funder statement in your cover letter; we will change the online submission form on your behalf. 6. Please provide a complete Data Availability Statement in the submission form, ensuring you include all necessary access information or a reason for why you are unable to make your data freely accessible. If your research concerns only data provided within your submission, please write "All data are in the manuscript and/or supporting information files" as your Data Availability Statement. 7. Your ethics statement should only appear in the Methods section of your manuscript. If your ethics statement is written in any section besides the Methods, please move it to the Methods section and delete it from any other section. Please ensure that your ethics statement is included in your manuscript, as the ethics statement entered into the online submission form will not be published alongside your manuscript.  8. Please include captions for your Supporting Information files at the end of your manuscript, and update any in-text citations to match accordingly. Please see our Supporting Information guidelines for more information: http://journals.plos.org/plosone/s/supporting-information.

Reviewers' comments:

Reviewer's Responses to Questions

**Comments to the Author**

1. Is the manuscript technically sound, and do the data support the conclusions?

Reviewer #1: Partly

Reviewer #2: Yes

2. Has the statistical analysis been performed appropriately and rigorously? 

Reviewer #1: N/A

Reviewer #2: Yes

3. Have the authors made all data underlying the findings in their manuscript fully available?

Reviewer #1: Yes

Reviewer #2: Yes

4. Is the manuscript presented in an intelligible fashion and written in standard English?

Reviewer #1: Yes

Reviewer #2: No

5. Review Comments to the Author

Reviewer #1: Thank you for the opportunity to review this well-written, interesting manuscript reporting a study on respectful maternity care in Rwanda. It is a thorough report of a qualitative part of a larger study. I have a couple of points for the authors to consider generally, and other more specific suggestions below.

Consider throughout the manuscript using the language ‘birth’ or ‘giving birth’ rather than ‘delivery’, as it aligns more with respectful language by acknowledging the role of the mother in giving birth. The themes are coherent but need to be supported throughout by more raw data (participant quotes). The manuscript needs to include a section on ethics (ethics approval, how you approached ethical issues, etc).

Introduction

P 8, Paragraph 1, line 3 – should read ‘ensures’ autonomy (rather than ensured)

Paragraph 2, line 2/3 – consider replacing ‘pleasant delivery’ with ‘positive birth experience’

Page 10, para 2 – maybe a subheading introducing the change from Background (Maternity care in Rwanda) to methodology.

First sentence – consider foregrounding the methodology (eg AI is a suitable research methodology to explore RMC as it focuses on….)

In addition – consider moving this section after the small literature review that comes next.

Page 11, top of page – subheading – literature review (or similar) – and put this section before the couple of paragraphs where you introduce AI. This means you have stated the gap before introducing the methodology, so has a more cohesive structure.

P 12, Methods

Para 1 – all qualitative research is contextual – so not sure you need to say this.

Para 2 – ‘quantitative survey of mothers’ (rather than ‘on’). Also, please state the overall design of the study here? Is it mixed-methods of some sort?

P 13, study setting

Para 1, please add if you needed to get each site ethics approval – and include a subsection that addresses ethics.

Sampling & data collection

Para 1, consider changing ‘mothers who self-reported to be respected’ to ‘mothers who self-reported feeling respected’.

P 14, can you please add a bit more about recruitment? How did you access the women, were they given time to consider between receiving the information and agreeing to participate?

Para 2, was there a reason for the time frame for interview? 12 hours seems very close to birth. Please give more information about field notes. Please give more information about the time frame. How did you recruit and conduct 30 interviews in 30 days?

Stages of AI

P 15, para 2 – the second sentence (beginning ‘pose questions’ sounds more like a direction about how to conduct AI, rather than an explanation. Consider rephrasing, ‘eg ‘questions are posed….’.

Para 3 – typo – ‘what is care were’ (should be ‘where’)

Data analysis

P 17, para 3 – please give more detail about the specific thematic analysis process employed and provide reference.

P 18, para 2 – where you say ‘relied solely on mothers’ narratives’ – does not capture how your own positions influenced the creation of themes. Please add a section on researcher reflexivity to address how researcher bias is acknowledged in qualitative research.

P 19

Paragraph 1 (under table). Consider identifying the subthemes - the sentence as it reads currently is a bit confusing. E.g 1) Appreciated care, with subthemes: compassionate care and emotional support; autonomy and self-determination; timely care; privacy and confidentiality preservation; and enabling environment.

P 20

When describing the elements of the theme, consider explaining that these were what the women experienced. For example – ‘participants expressed examples of feeling treated with empathy and compassion. These included: receiving anaesthesia before suturing vaginal tears and episiotomy; being provided with painkillers after caesarean birth; and offered support to manage labour pain.’

P 22

From my perspective, being told when to push (directed pushing) and being advised to ‘practice self-control’ do not sound overly supportive. I understand I may be misinterpreting this, but perhaps it needs more explanation as to how / why this was experienced as positive, respectful care.

P 23

"Doctor always asked me " Would you come for a check-up to see your labour progress? Would you come for an ultrasound to check if the baby is still alive? and I said yes. They told me that I am going to have surgery, they told me that you have to sign and your companion needs to sign too; once you have signed, you have agreed to the surgery they are going to do for you". Participant 13, aged 28, gravida 1, C/s delivery

This does not sound to me much like informed consent. But I can see that the participant is happy to be asked (rather than told)? And perhaps it is lost in translation.

Maybe explain the context a bit more – it made more sense after I read the next paragraph, about being happy simply because they were not being coerced, and there being a level of expectation that consent would not happen.

P 24

The text talks about an experienced midwife but the quote is about the doctor.

Also, there are quite a few claims throughout that do not have the support of raw data – e.g. p 23/24 “Most of the mothers expressed happiness and were impressed by the care received. They reported receiving the best services and all the necessary care. The HCP attended to them promptly and assisted them every time, even when they were working overtime. In some hospitals, they first use available medicines, and mothers reimburse them later. The entire process was well organized, and mothers who arrived in critical conditions were given hope and excellent care.”

It would be good to see these points supported by more participant quotes.

Para 2 – “Privacy was ensured as described by the mothers” This describes how they felt they were treated, but I wonder if it would be clearer if these were first stated as important aspects of RMC – eg the participants described privacy as an important part of feeling well cared for.

Final para – in the quote - the word ‘colleague’ usually refers to a person you work with – maybe think of another word to describe what they mean here – the other women in the hospital.

P 25 Enabling environment

Para 1 – please explain context a little more. Who are the birth companions usually? They seem to have a pre-determined role. Are they family members?

P 29

The subtheme of Mother self-control does not seem to be answering the question, which is about their experience of receiving RMC. Arguably RMC should occur no matter how many babies a woman has. I wonder is there is a way to reconsider this – acknowledging the participants appeared to take responsibility for how they were treated (or expected to be treated). Provider behaviours subtheme is much more relevant and rich – maybe focus more on that.

P 31

First quote – typo “Why didn't you give me my [missing word] off?"

Again, this section would benefit from the inclusion of more participant quotes.

P 33 Discussion

Para 2, line 6 – Consider rephrasing the sentence beginning ‘It was impressive…’ mainly because as language reporting on research it is too emotive. Perhaps ‘Interestingly…’

Para 3, line 2 – ‘at some health facility’ should read ‘at one health facility’

P 34, para 2, line 3 missing word ‘however, consent [for] care’

Para 3 – this explanation of family & birth companions is helpful – perhaps mention this earlier in the introduction.

P 35 Conclusions

Para 1 - As well as saying what ‘should’ be done in terms of RMC, first briefly summarise the themes, as what the participants said they experienced as RMC.

Para 2 – what specifically need to be documented? How is RMC delivered? How are people trained? How many facilities offer training?

Reviewer #2: The research primarily explores how mothers in Rwanda's Eastern Province perceive and appreciate their childbirth experiences, specifically focusing on Respectful Maternity Care (RMC). This question is highly relevant, as understanding positive RMC experiences can offer valuable insights into effective practices and culturally appropriate solutions, particularly in resource-limited settings. The focus on positive experiences through Appreciative Inquiry (AI) adds a unique dimension to the existing literature, which often emphasizes negative aspects like disrespect and abuse.

See the uploaded reviewer comment attached

6. PLOS authors have the option to publish the peer review history of their article (what does this mean?). If published, this will include your full peer review and any attached files.

Reviewer #1: No

Reviewer #2: No

---

## [Editor Report · Decision Letter 1]

27 Nov 2024

Experiences and perceptions of respectful maternity care among mothers during childbirth in health facilities of Eastern province of Rwanda: An Appreciative Inquiry

PONE-D-24-26291R1

Dear Dr. Muhayimana,

We’re pleased to inform you that your manuscript has been judged scientifically suitable for publication and will be formally accepted for publication once it meets all outstanding technical requirements.

Kind regards,

Mukhtiar Baig, Ph.D.

Academic Editor

PLOS ONE

---

## [Editor Report · Acceptance letter]

7 Jan 2025

PONE-D-24-26291R1 

PLOS ONE

Dear Dr. Muhayimana, 

I'm pleased to inform you that your manuscript has been deemed suitable for publication in PLOS ONE. Congratulations! Your manuscript is now being handed over to our production team.

Kind regards, 

on behalf of

Professor Mukhtiar Baig 

Academic Editor

PLOS ONE